# Interplay between Vitamin D and Adipose Tissue: Implications for Adipogenesis and Adipose Tissue Function

**DOI:** 10.3390/nu15224832

**Published:** 2023-11-18

**Authors:** Shiqi Lu, Zhen-Bo Cao

**Affiliations:** School of Exercise and Health, Shanghai University of Sport, Shanghai 200438, China; luss_77@163.com

**Keywords:** vitamin D, adipose tissue, adipogenesis, inflammatory response, obesity treatment, VD/VDR pathway

## Abstract

Adipose tissue encompasses various types, including White Adipose Tissue (WAT), Brown Adipose Tissue (BAT), and beige adipose tissue, each having distinct roles in energy storage and thermogenesis. Vitamin D (VD), a fat-soluble vitamin, maintains a complex interplay with adipose tissue, exerting significant effects through its receptor (VDR) on the normal development and functioning of adipocytes. The VDR and associated metabolic enzymes are widely expressed in the adipocytes of both rodents and humans, and they partake in the regulation of fat metabolism and functionality through various pathways. These encompass adipocyte differentiation, adipogenesis, inflammatory responses, and adipokine synthesis and secretion. This review primarily appraises the role and mechanisms of VD in different adipocyte differentiation, lipid formation, and inflammatory responses, concentrating on the pivotal role of the VD/VDR pathway in adipogenesis. This insight furnishes new perspectives for the development of micronutrient-related intervention strategies in the prevention and treatment of obesity.

## 1. Introduction

Adipose tissue constitutes a heterogenous organ, comprising distinct subtypes including WAT, beige adipose tissue, and BAT [1,2,3]. These adipose depots exhibit dissimilar morphological structures and physiological functions. WAT, being the largest lipid storage site within the body, chiefly constitutes white adipocytes [4]. Typical white adipocytes possess a singular large lipid droplet and a limited number of mitochondria, primarily involved in energy storage and provision. Impaired functionality of white adipocytes can result in disrupted lipid metabolism, lipodystrophy, and insulin resistance (IR) due to lipid spillage [5,6]. Moreover, excessive accumulation of WAT is associated with obesity, adversely impacting various physiological processes, and represents a prominent risk factor for the onset of type 2 diabetes and cardiovascular diseases [7]. Brown adipocytes exhibit a distinct phenotype characterized by the presence of multilocular small lipid droplets and a higher density of mitochondria [8]. These cells are prominently marked by elevated expression levels of mitochondrial uncoupling protein 1 (UCP1), which facilitates mitochondrial heat generation via the uncoupling of oxidative phosphorylation, thus ensuring the maintenance of basal body temperature in cold environments instead of relying on shivering [9]. Beige adipose tissue, displaying similar morphological and functional characteristics to BAT, is perceived as an intermediary state between WAT and BAT. The emergence of beige fat typically occurs after birth and is often juxtaposed with white adipocytes that transition between a browning or whitening state in response to environmental stimuli [4,8,10]. The induction of beige adipogenesis in WAT has been proposed as a potential therapeutic strategy for combating obesity and metabolic disorders associated with excessive weight gain.

Numerous investigations have established a close association between the concentration of vitamin D (VD) in systemic circulation and body adiposity [11]. Observational studies have consistently demonstrated a negative correlation between the serum levels of 25-hydroxyvitamin D (25(OH)D) and 1,25-dihydroxyvitamin D3 (1,25(OH)_2_D_3_), the bioactive form of VD, with measures of body mass index (BMI), subcutaneous fat mass, and visceral fat mass [12,13,14]. Multiple hypotheses have been posited to elucidate the underlying mechanisms connecting obesity with low serum 25(OH)D levels, encompassing volumetric dilution, adipose tissue sequestration, metabolic disturbances, and inadequate exposure to sunlight [15,16,17]. Mounting evidence lends support to the adipose tissue sequestration hypothesis, as research has consistently revealed that a high-fat diet augments the expression of Cyp2r1 in adipose tissue, facilitating the active uptake of vitamin D3 (cholecalciferol) and its subsequent conversion into 25(OH)D for storage within lipid droplets [18]. Furthermore, investigations have revealed that individuals with metabolically unhealthy obesity, which primarily involves the accrual of visceral fat, exhibit lower levels of 25(OH)D compared to those with metabolically healthy obesity, characterized by subcutaneous fat accumulation [19]. This observation implies that diverse loci of visceral fat deposition could exert varying effects on VD concentrations, thereby necessitating an exploration into the potential role of this factor in mediating metabolic dysregulation.

Moreover, the metabolic regulation of VD in adipose tissue is intricately linked to the VD receptor (VDR). Substantial evidence demonstrates widespread expression of VD-related metabolic enzymes and the VDR in human adipose tissue [20]. Interaction between the VDR and 1,25(OH)_2_D_3_ enables the direct or indirect modulation of the local VD metabolism as well as the regulation of adipocyte differentiation, inflammation, and other critical processes [20,21,22]. Interestingly, investigations have revealed higher expression levels of the VDR in the visceral adipose tissue (VAT) of obese individuals compared to lean subjects, while no significant differences were observed in the subcutaneous adipose tissue (SAT) [17]. Furthermore, in healthy non-obese individuals, elevated serum 25(OH)D levels were significantly associated with reduced VDR expression specifically in VAT, whereas this correlation was absent in SAT [23]. This reinforces the notion that VD may exert distinct effects on adipose tissue at different anatomical sites.

Therefore, this review focuses on the functional differences of VD in different types of adipocytes and provides new intervention ideas of micronutrients for the prevention and treatment of obesity.

## 2. Vitamin D Metabolism and Metabolic Health

Nature encompasses two primary forms of VD, namely vitamin D2 (ergocalciferol) and vitamin D3 [24]. Vitamin D2 predominantly originates from plant-based sources like mushrooms, while vitamin D3 primarily derives from cutaneous synthesis in response to ultraviolet (UV) light exposure and animal-based foods such as egg yolk, cod liver oil, sardines, and milk [25,26]. Both dietary vitamin D2 and vitamin D3 are absorbed into the bloodstream through chylomicron transport within the small intestine. Conversely, endogenous VD synthesis predominantly occurs through the conversion of 7-dehydrocholesterol in the skin, facilitated by exposure to UV light (290–315 nm). The process begins with the rapid conversion of a certain substance to vitamin D3, which is facilitated by heat. This newly formed vitamin D3 then binds to a specific protein known as vitamin D-binding protein (VDBP). Once bound, VDBP enters the circulatory system and is either transported to the liver or stored in fat along with the circulating blood [27]. Vitamin D3 may be further metabolized in the liver, enriched with hydroxylase enzymes like CYP2R1, resulting in the formation of 25(OH)D, which serves as the clinically utilized form for assessing VD status [28,29]. By the criteria established by the Endocrine Society, a serum concentration of 25 (OH) D ranging from 30 to 100 ng/mL is considered within the normal range. A concentration below 30 ng/mL (75 nmol/L) indicates insufficiency of VD, while a concentration below 20 ng/mL (50 nmol/L) indicates deficiency. Insufficient levels of VD can have detrimental effects on bone health, leading to fractures and bone loss [30,31]. Severe vitamin D deficiency (VDD) is defined by a serum 25 (OH) D concentration below 12 ng/mL (30 nmol/L), which significantly elevates the risk of mortality, immune system disorders, and other diseases [32].

The conversion of 25 (OH) D to its active form, 1,25 (OH) _2_D_3_, occurs in the kidney through the enzymatic action of 1alpha hydroxylase (CYP27B1). This conversion facilitates the exertion of biological effects by 1,25 (OH) _2_D_3_. These effects can be categorized into three main mechanisms:

(1) Genomic effect: The formed 1,25 (OH) _2_D_3_-Vitamin D receptor (VDR)-Retinoid X Acid Receptor (RXR) complex translocates from the cytoplasm to the nucleus, where it binds to the VD response element (VDRE) on target genes. Consequently, the complex regulates the expression of numerous genes, totaling in the hundreds [33].

(2) Non-genomic effect: the membrane VDR activation initiates a rapid membrane priming reaction, triggering signaling pathways that contribute to biological responses [34,35].

(3) Epigenetic effects: these effects involve the regulation of miRNAs, DNA methylation, histone acetylation/deacetylation, and histone methylation/demethylation, which collectively impact gene expression regulation.

Ultimately, the elevated levels of 1,25 (OH) _2_D_3_ induce the action of CYP24A1-mediated 24-hydroxylation. This process breaks down 1,25 (OH) _2_D_3_ into biologically inactive calcific acid, which is ultimately excreted with bile, thereby completing the overall metabolic process [29].

VD exerts a crucial regulatory role in substance metabolism and displays significant potential in non-bone-related metabolic processes [36]. An increasing body of evidence has established a close association between circulating 25 (OH) D levels and the development and progression of cardiovascular disease, type 2 diabetes, IR, and obesity [14,37,38]. Investigations have revealed that in adipocytes, diet-induced VDD contributes to elevated levels of macrophage infiltration and inflammation within rat adipose tissue [39]. Furthermore, VD insufficiency reduces the activity of sirtuin1 (STRT1) and adenosine monophosphate-activated protein kinase (AMPK), which consequently influences energy metabolism and the occurrence of inflammatory responses [21].

Regarding the anti-inflammatory effects of VD on adipose tissue, consistent findings have been obtained from in vitro and in vivo studies. In both preadipocytes and adipocytes, 1,25 (OH) _2_D_3_ inhibits the expression of IL-6, IL-1β, IL-8, MCP-1, and leptin while stimulating the expression of adiponectin [40,41,42,43]. Dietary VD supplementation also significantly attenuates the expression of chemokines and macrophage infiltration in mouse adipose tissue [43]. Cumulatively, the adverse effects of VD on skeletal muscle and adipose tissue contribute to the development and progression of IR and obesity.

## 3. Adipose Tissue Function and Metabolic Health

Adipose tissue represents a heterogeneous organ characterized by substantial plasticity, encompassing diverse cell types including preadipocytes, immune cells, and mesenchymal stem cells [44]. Throughout its life cycle, adipose tissue undergoes a dynamic process of expansion and compression in response to shifting metabolic demands. A higher adipose tissue mass generally correlates with compromised metabolic health. However, adipose tissue expansion serves as a mechanism to safely store surplus nutrients and prevent their accumulation in other tissues. Nevertheless, excessive adipose tissue undermines metabolic health, whereas insufficient adipose tissue, as observed in lipodystrophy, can lead to metabolic disorders such as diabetes, hypertriglyceridemia, non-alcoholic fatty liver disease, and adipose tissue-related endocrine dysfunctions [45]. Hence, the proper functioning of adipose tissue is crucial for maintaining metabolic well-being.

Adipose tissue plays a critical role in the biosynthesis and breakdown of free fatty acids and other nutrients. It further modulates the metabolic functions of other organs through the secretion of adipokines, including adiponectin and leptin [46]. Hence, the preservation of a proper equilibrium among adipocyte hypertrophy, proliferation, fibrosis, and lipolysis, along with the normal synthesis and release of adipokines, holds paramount importance for metabolic well-being. However, it is noteworthy that distinct types of adipose tissue exhibit divergent contributions to metabolic health.

### 3.1. White Adipose Tissue

WAT is a prominent organ within the human body, exhibiting a significant capacity for storing and releasing energy in the form of triglycerides. Furthermore, WAT secretes adipokines in response to variations in systemic energy levels [47,48]. In normal circumstances, WAT adapts to increased lipid storage demands by undergoing adipocyte expansion, specifically hyperplasia, and hypertrophy, to maintain a consistent energy supply during periods of food deprivation. However, aberrations in one’s lifestyle and dietary composition have given rise to chronic overnutrition, leading to the surpassing of the tolerable threshold for stored triglycerides. Consequently, excess triglycerides are transferred to other metabolic organs, such as the muscle and liver, resulting in ectopic lipid accumulation. This accumulation plays a mediating role in the development of IR and complications related to obesity [3,49].

Based on anatomy, WAT can be subdivided into VAT and SAT [50]. The ratio between these two depots holds substantial implications for metabolic health. Metabolically unhealthy individuals with obesity exhibit a heightened visceral adiposity index (a specific index incorporating waist circumference, triglycerides, and HDL to indirectly represent VAT function) and a homeostatic model assessment of insulin resistance (HOMA-IR), ultimately resulting in poorer metabolic parameters when compared to their metabolically healthy obese counterparts [51,52]. Notably, metabolically healthy individuals with obesity possess lower amounts of visceral fat, whereas metabolically unhealthy individuals with normal weight exhibit higher levels of visceral fat [53]. This evidence strongly suggests that increased VAT significantly correlates with more severe IR and metabolic diseases associated with obesity, even in individuals of normal weight [54]. This correlation can be attributed to VAT’s propensity for lipolysis and immune cell infiltration, consequently resulting in the production of proinflammatory factors such as TNF-α, IL-6, and IL-1β, which contribute to IR. Nonetheless, these processes also serve as critical means of internal organ protection [55,56,57,58]. The distinction between VAT and SAT is further evident in the volume of adipocytes, which tends to be larger in VAT and smaller in SAT. As a consequence, hypertrophic adipocytes manifest IR [59,60]. This disparity is also responsible for explaining why obesity characterized by SAT accumulation is comparatively healthier than that dominated by VAT [61].

### 3.2. Brown Adipose Tissue and Beige Adipose Tissue

In the human body, BAT has relatively little quantity and is selectively distributed in specific anatomical parts, such as the neck and scapula [62]. BAT possesses distinct characteristics, including a darker appearance, multiloculated lipid droplets, and a greater mitochondrial density within individual brown adipocytes, which play a crucial role in maintaining lipolysis homeostasis by promoting fat burning [8,63]. During infancy, the thermogenic effect of BAT largely determines cold resistance. By employing uncoupled oxidative phosphorylation rather than shivering, BAT generates heat to sustain the basal body temperature of organisms in cold environments [9]. Recent investigations have revealed that the significance of BAT extends beyond thermogenesis and body temperature regulation, encompassing important implications in the fine-tuning of glucose and lipid metabolism as well as insulin sensitivity [64]. Activation of BAT can substantially enhance whole-body energy expenditure by more than 100% in mice and 40–80% in humans, concurrently reducing plasma triglyceride levels [65,66]. However, the content of BAT gradually diminishes with age in humans [67]. Consequently, the activation of adult BAT holds considerable importance in maintaining metabolic health. Additionally, researchers have identified a special type of “hypothermogenic” beige adipocytes, which possess larger lipid droplets, fewer mitochondria, and reduced thermogenic gene expression levels [68]. Beige adipose tissue represents an intermediary state between WAT and BAT [8,10]. Beige adipocytes can transform mature white adipocytes, a plastic process primarily occurring within SAT [69,70,71]. Cold exposure, catecholamines, exercise, adipokines, and other stimuli can induce the browning or beginning of WAT [72,73,74,75]. However, obesity may impede the thermogenic function of BAT, leading to a significant reduction in BAT activity among obese individuals, irrespective of their BMI. Notably, this reduction in BAT activity exhibits a strong negative correlation with VAT mass [76,77]. While similar results were not observed in the BAT of diet-induced obese mice, ob/ob mice exhibited a decrease in the mRNA expression levels of UCP1 in their BAT [78].

## 4. Vitamin D Plays an Important Role in Adipose Tissue

The relationship between VD and adipose tissue is intricate. Adipose tissue, particularly VAT, serves as a primary reservoir for VD, with approximately 65% of the whole-body vitamin D3 and 35% of 25(OH)D being stored in lipid droplets within adipose tissue [79,80,81]. Conversely, adipocytes express VD-related metabolic enzymes and VDR, which in turn regulate the local VD metabolism, adipogenesis, lipid metabolism, thermogenesis, inflammation, and apoptosis through direct or indirect interactions with 1,25(OH)_2_D_3_ [4,29].

### 4.1. Vitamin D and Visceral Adipose Tissue

VAT has been observed to contain approximately 20% higher levels of VD compared to SAT, although the precise reasons for this difference remain unclear [81]. Serum VD levels show a negative correlation with VAT content and serve as an important indicator for assessing the expression of the VDR within VAT [74,82,83]. While the precise relationship between the VDR in VAT and serum 1,25(OH)_2_D_3_ has yet to be fully elucidated, the current findings provide support for a dose–response relationship between the severity of VDD and low VDR expression [84]. Notably, VDR expression in VAT is significantly lower in individuals with a normal BMI compared to those who are obese (BMI = 30–40 kg/m^2^) or severely obese (BMI > 40 kg/m^2^). Several studies have found a positive correlation between waist circumference and VDR expression in VAT, as well as a negative correlation between 25(OH)D and VDR expression in VAT [23]. These results align with findings from animal studies involving VDR overexpression, suggesting that the VDR plays a role in the proliferation or hypertrophy of visceral adipocytes and is closely associated with the development of central obesity.

Several recent experimental and clinical investigations have elucidated the anti-inflammatory properties of VD in the liver, muscle, and adipose tissue, in which the inhibition of monocyte chemoattractant protein 1 (MCP-1) and macrophage recruitment has been observed [4,85,86]. Hypomethylation of inflammation-related adipokines (BCL5, CXCL8, IL-12A) has been observed in the adipose tissue of individuals with VDD, particularly in obese subjects. This association is linked to an increase in total fat mass, visceral fat mass, and impaired insulin sensitivity [87,88]. In the study conducted by Imaduddin Mirza et al., it was found that macrophage infiltration in the VAT of obese individuals demonstrated a dose-dependent increase with the increasing VDD [88]. Conversely, restoring VD status has been shown to improve the inflammatory response observed in the VAT of obese individuals [89]. Animal studies have reported similar findings, observing that diet-induced VDD exaggerates adipocyte hypertrophy and recruits adipose tissue macrophages to the epididymal adipose tissue, resulting in elevated levels of IL-6 and TNF-α [21,39]. Furthermore, VDD has been associated with reduced STRT1 and AMPK activity, affecting energy metabolism and triggering inflammatory responses [21]. In mice subjected to 12 weeks of VD restriction, a significant increase in NF-kB levels was observed in the visceral fat [39]. Moreover, female mice with fat-specific knockdown of the VDR showed an increase in VAT mass, resembling the VDD phenotype [90]. Interestingly, Matthews et al. found that in the context of high-fat feeding, mice with fat-specific VDR knockout exhibited higher serum 25 (OH) D levels compared to control mice, with no significant differences observed in the visceral adipocyte area [90]. This finding may be attributed to the upregulation of thermogenic proteins like UCP1, which promote lipolysis in VAT. Furthermore, studies have indicated that VD supplementation restricts macrophage migration in the adipocytes of obese mice, thereby exerting an anti-inflammatory effect [43]. Collectively, these findings underscore the crucial involvement of VD in VAT accumulation and inflammation, suggesting that it may serve as a significant regulatory factor in the pathogenesis of central obesity. Moreover, the VD/VDR complex may play a role in modulating the NLRP3/caspase 1 pathway, though the underlying mechanisms in adipocytes remain unknown [91]. Further investigations are warranted to substantiate the suggested connections between the NLRP3 inflammasome, obesity, IR, and VD, which may shed light on the potential role of VD in attenuating IR and metabolic disorders.

### 4.2. Vitamin D and Subcutaneous Adipose Tissue

VDD has been observed to promote SAT accumulation in the forearm, arm, and thigh regions of the human body, similar to its effect on VAT [92]. Animal studies have consistently demonstrated an increase in SAT in high-fat diet-induced VDD animals [21]. However, some studies have reported a negative correlation between serum 25 (OH) D levels and body fat, but no significant association between serum 25 (OH) D and SAT [93]. In the absence of differences in food and energy intake, high-fat diet-fed VDD animals exhibited significantly higher epididymal fat deposition compared to high-fat diet-fed VD-sufficient animals, while the SAT did not differ between the two groups [94,95]. This finding suggests that VDD may preferentially facilitate lipid accumulation in VAT rather than SAT under high-fat diet conditions. Interestingly, BMI and HOMA-IR were found to be positive predictors of VDR expression in SAT [23]. The response of SAT to 1,25 (OH)_2_D_3_ treatment and the expression of the VDR vary depending on the degree of obesity [23,96]. Moreover, in obese male patients, it has been noted that isoproterenol-mediated lipolysis in abdominal SAT is attenuated, leading to a reduction in the release of 1,25(OH)_2_D_3_ [97]. However, a 12-week period of energy restriction and body fat loss did not result in significant changes in the 25 (OH) D level in the SAT or serum 25 (OH) D concentration [98]. This suggests that weight loss may trigger the release of 25 (OH) D in SAT, but the magnitude of release may not be sufficient to cause noticeable changes in serum 25 (OH) D. Alternatively, the released 25 (OH) D may undergo inactivation and subsequent excretion through the classical negative feedback regulation pathway. Notably, the expression of CYP2J2, CYP27A1 (25-hydroxylase), and CYP27B1 (1α-hydroxylase) in SAT is lower in obese individuals, whereas the expression level of CYP24A1 was significantly increased after weight loss [17]. This supports the notion that VD inactivation is heightened in SAT following weight loss.

VD also plays a crucial role in the inflammatory processes occurring in SAT. In a study, SAT samples obtained from obese subjects were manipulated in vitro with IL-1β intervention to induce an inflammatory model. Following incubation with 1,25 (OH)_2_D_3_, the mRNA levels of IL-6, IL-8, and MCP-1 in the cells were significantly decreased [99]. Although this result was not observed in SAT samples from subjects who orally consumed VD supplements, it is important to consider that obesity can impair the effectiveness of VD supplementation. Furthermore, the results from in vitro studies underscore the close association between the anti-inflammatory effect of VD and changes in the VDR in tissues. Notably, it has been demonstrated that upregulated VDR mRNA and downregulated CYP27B1 mRNA are positively correlated with the expression levels of IL-1β, IL-6, and IL-8 mRNA in SAT samples from obese subjects [100]. However, it should be noted that some studies have shown that VDR expression is higher in VAT but not in SAT in obese patients [17]. In non-obese subjects, a clear negative correlation was observed between serum VD levels and VDR expression in VAT, but this phenomenon was not observed in SAT [23]. These findings suggest that VD metabolism and VDR expression in SAT are relatively stable, likely due to SAT’s higher stability and anti-lipolytic effect. With the progression of obesity, metabolic changes associated with VD may initially occur in VAT rather than SAT. However, further research is needed to elucidate the underlying mechanisms.

### 4.3. VDR Regulates Adipose Tissue Browning and Alters Energy Expenditure

VDR expression levels have a significant impact on the quality of BAT. Studies have demonstrated that mice with fat-specific VDR overexpression exhibit a considerably higher BAT content compared to control mice, whereas fat-specific VDR knockout and whole-body VDR knockout mice have a lower BAT content [101,102]. The activity of uncoupling proteins (UCPs) is directly modulated by VD/VDR signaling, and the dose-dependent inhibition of brown adipocyte differentiation by 1,25 (OH)_2_D_3_ has been observed [78,103]. Primary BAT culture experiments have confirmed the ability of 1,25 (OH)2D_3_ to directly inhibit UCP expression [104]. VDR overexpression suppresses brown adipocyte differentiation and peroxisome proliferator-activated receptor gamma (PPAR-γ) activation by downregulating peroxisome proliferator-activated receptor gamma coactivator 1 alpha (PGC-1α) and PR domain containing 16 (PRDM16), subsequently reducing the expression levels of UCP1, UCP2, and UCP3 in BAT. These effects are reversed upon VDR knockdown [78,102,105]. Both VDR knockout and 1α-hydroxylase (CYP27B1) knockout mice exhibit weight loss and increased UCP1 expression in BAT [104]. Similarly, the UCP1 gene expression was significantly upregulated in the VAT of mice with fat-specific VDR knockout, leading to increased energy expenditure and promotion of the browning process in WAT [90]. Furthermore, VD supplementation has been shown to inhibit the browning of WAT in mice with chronic kidney disease [106]. Collectively, these findings indicate that the VDR serves as a negative regulator of fat browning.

UCP2, a member of the UCPs family, occupies a pivotal position in the regulation of energy metabolism. Previous investigations have revealed that the administration of low doses of VD can suppress Ucp2 expression by bolstering mitochondrial localization and ATP generation, thus restraining the initiation of apoptosis [107]. Conversely, high doses of VD can trigger this process. In UCP2-transfected 3T3-L1 cells, 1,25(OH)_2_D_3_ displayed the dose-dependent stimulation of cytosolic Ca^2+^ levels, leading to a 25% increase in mitochondrial Ca^2+^ levels, thereby further propelling adipocyte apoptosis [108]. These results are consistent with the observation of lean phenotype and excessive energy expenditure in VDR knockout mice. It is postulated that VDR-null mice undergo adipocyte apoptosis, resulting in insufficient adipose tissue to maintain body temperature, prompting augmented UCPs expression in BAT and WAT to generate heat. Nevertheless, VD may mitigate this phenomenon by stimulating the differentiation of MSCs into preadipocytes, which in turn inhibits the expression of UCPs. It is imperative to further explore the role of the 1,25(OH)_2_D_3_/VDR-UCPs pathway system in the regulation of BAT development and function in obesity.

### 4.4. Vitamin D Supplementation Regulates Adipose Tissue Metabolic Health

VDD plays a pivotal role in the pathogenesis and progression of metabolic disorders, such as IR. To maintain serum 25 (OH) D levels within the recommended range of 20 to 100 ng/mL, the Endocrine Society recommends a daily intake of 1500 to 2000 IU or higher of VD to mitigate the risk of non-skeletal diseases [25]. Nevertheless, the reproducibility of the effect of VD supplementation in improving serum 25 (OH) D levels is limited, particularly in the context of obesity [109,110]. It is noteworthy that systematic reviews and meta-analyses have reported a significant increase in serum leptin levels and a decrease in the levels of C-reactive protein in individuals with type 2 diabetes mellitus following VD supplementation. These findings are beneficial for ameliorating systemic inflammation [111,112]. Therefore, some studies have suggested that the efficacy of VD supplementation may be impaired in the presence of obesity, and the beneficial effects of supplementation may only be observable in individuals with inadequate VD levels or severe disorders of glucose and lipid metabolism [110]. Hence, changes in serum 25 (OH) D levels after VD supplementation should be carefully considered.

In a trial conducted on obese Wistar rats, supplementation with 800 IU of VD demonstrated a remarkable ability to attenuate weight gain and decrease the deposition of abdominal fat [113]. Consistent with these findings, obese Wistar rats receiving a higher dosage of 2400 IU of VD exhibited reduced levels of VAT leptin and MCP-1 mRNA, along with elevated levels of adiponectin, as compared to obese rats supplemented with either 800 IU or placebo [114]. However, the study did not ascertain any variations in the serum 25 (OH) D levels pre- and post-vitamin D supplementation, thereby making it inconclusive to determine whether the favorable effects of VD supplementation on obesity can be attributed to changes in VD levels.

The absence of any observable changes in the physiological morphology of adipocytes following VD supplementation has led to the suggestion that it may not possess beneficial effects on the development of obesity. Nonetheless, it should be noted that certain studies have demonstrated significant improvements in adipose tissue inflammation and reduction in liver tissue steatosis in obese C57BL/6J mice subjected to VD supplementation [115,116]. In diet-induced obese mice (an animal model of metabolic inflammation) and mice injected intraperitoneally with lipopolysaccharide (an animal model of acute inflammation), VD has been shown to decrease the levels of proinflammatory cytokines and chemokines in both adipocytes and VAT [43]. Furthermore, VD supplementation has been found to lower the mRNA expression levels of Cyp27A1, Cyp24A1, and cubilin in epidydimal white adipose tissue (eWAT), indicating a possible modulation of its metabolism through reducing the uptake and activation of 25 (OH) D [117]. Glucose transport in the adipose tissue of high-fat diet-fed mice also increased after VD supplementation, suggesting a potential improvement in obesity-induced glucose metabolism disorders [118]. Notably, VD in conjunction with high calcium intake can activate calcium-mediated apoptotic pathways in adipose tissue, activating calcium-dependent apoptotic proteases calpain and caspase-12, which presents a potential avenue for obesity prevention and treatment [119]. However, the reliability and controllability of this approach warrant further investigation, ensuring that healthy cells are not compelled to undergo apoptosis while promoting the apoptosis of dysfunctional adipocytes to curb inflammatory infiltration arising from hypertrophy.

## 5. VDR Plays a Key Role in the Regulation of Adipogenesis

VDR plays a crucial role in adipogenesis. Systemic VDR-null mice display diminished adipose tissue mass, augmented overall energy expenditure, and resistance to high-fat diet-induced obesity, which coincides with notable enhancements in glucose tolerance and insulin sensitivity [102,120]. Furthermore, these defects in cellular adipogenesis become more evident as the mice age [101]. Similarly, another study found no significant difference in VAT cell size between VDR knockout mice and wild-type mice at 21 days of age. However, as the mice reached 8 months of age, VDR knockout mice exhibited significantly smaller fat cell areas compared to wild-type mice, accompanied by alopecia and increased energy expenditure [121]. Importantly, no disparity in VAT cell size was observed between VDR knockout mice at 21 days and 8 months [121]. In alignment with the lean phenotype observed in VDR knockout mice, VDR-deficient stem cells exhibited impaired adipogenesis in the presence and absence of 1,25(OH)_2_D_3_, and they hindered 1,25(OH)_2_D_3_-mediated lipogenesis in human adipogenic progenitor cells when a VDR antagonist was present. These findings indicate the vital role of the VDR in adipogenesis [122]. In a separate in vitro investigation, VDR knockdown was observed to hinder adipogenesis in 3T3-L1 cells, while 1,25(OH)_2_D_3_ demonstrated the ability to enhance the differentiation of human and mouse adipose tissue-derived stem cells (ASCs) into adipocytes [34,123]. Mesenchymal cells sourced from 6-month-old VDR-null mice displayed impeded adipogenesis in vitro; however, their differentiation capabilities were restored upon the stable expression of the VDR [122].

However, deletion specific to the VDR in adipose tissue did not fully replicate the lean phenotype observed in mice with a complete knockout of the VDR gene throughout the entire body. Adipose tissue-specific VDR knockout mice showed an increase in VAT weight, while SAT accumulation remained resistant to a high-fat diet [90]. This indicates that the VDR responds differently to the physiological regulation of VAT and SAT. Interestingly, in the absence of significant differences in food intake, mice with a VDR knockout specific to mature adipocytes exhibited a higher fat mass and increased serum leptin levels when exposed to the same high-fat diet as mice with a complete knockout of the VDR gene throughout the entire body [90]. This suggests that the lean phenotype and resistance to a high-fat diet resulting from whole-body VDR knockdown are not solely due to the loss of VDR action in mature adipocytes. The authors of this study discovered that even with a specific VDR knockdown, adipose tissue still exhibited a weak VDR expression, highlighting the possibility that the role of the VDR in MSCs and preadipocytes may be the underlying factor responsible for this difference. However, it is important to note that the PPAR-γ gene expression in VAT was significantly increased in both the total VDR knockout and adipose tissue-specific VDR knockout mice, whereas other genes related to adipogenesis, such as Steap4, Esr, Dok1, and Acaca, did not show any changes due to genotype differences [90]. This could be attributed to the fact that similar to the VDR, PPAR-γ also requires binding to RXR-α, and in the absence of the VDR, more PPAR-γ may bind to RXR-α to exert its effects.

Generalized or adipose tissue-specific overexpression of the VDR in mice results in fat accumulation and elevated leptin levels [102,119,124]. Specifically, overexpression of the VDR in mouse adipose tissue leads to increased body weight and fat mass, disruption of glucose and lipid metabolism, development of IR, and impairment of thermoregulation [102,125,126]. Similarly, VDR overexpression in adipocytes increases adipose tissue mass and lipid storage [122]. However, contrary to the phenotype observed in VDR overexpressing mice, multiple in vitro studies demonstrate that 1,25(OH)_2_D_3_, an active form of VD, has an inhibitory effect on lipid accumulation in mature adipocytes. 1,25(OH)_2_D_3_ reduces triglyceride (TAG) accumulation by enhancing basal and adrenergic-stimulated lipolysis, while also decreasing de novo lipogenesis in 3T3-L1 adipocytes [22]. It also stimulates the mRNA expression and fatty acid (FA) oxidation rate of FA oxidation-related genes, including CPT1A, PGC-1α, PPAR-α, and UCP1 in 3T3-L1 adipocytes [22,127]. Moreover, 1,25(OH)_2_D_3_ enhances insulin-stimulated AKT phosphorylation, GLUT4 translocation, and glucose transport in 3T3-L1 adipocytes [40,118]. It is worth noting that in the presence of an adipogenic medium, 1,25(OH)_2_D_3_ promotes lipid accumulation and upregulates the expression of genes involved in lipid synthesis, such as FABP4, FASN, and PPAR-γ [122]. These findings indicate that 1,25(OH)_2_D_3_ exerts a significant influence on lipolysis and lipid accumulation in mature adipocytes.

The disparities between in vitro and in vivo outcomes can potentially be attributed to divergent VD/VDR functions during various phases of cell differentiation. In this context, the VDR plays a pivotal role in facilitating the maturation of MSCs into fully developed adipocytes, thus augmenting adipocyte proliferation to enable the efficient storage of nutrients in a comparatively healthful state. Conversely, insufficiency of the VDR impedes the emergence of new, mature white adipocytes, initiates the browning process, and amplifies energy expenditure. “Although this might prove advantageous for combating obesity by promoting fat loss, it is imperative to maintain a sufficient population of white adipocytes for the sake of optimal metabolic function”. Taking an evolutionary standpoint, such a scenario is disadvantageous in terms of securing the adequate energy reserves needed to withstand periods of food scarcity.

It is important to consider that recent epidemiological studies have provided substantial evidence highlighting the close correlation between maternal VD status and the development of obesity in offspring. Maternal serum 25 (OH) D levels during pregnancy exert an influence on various adulthood obesity-related parameters, such as BMI, body fat percentage, and waist circumference, consequently increasing the susceptibility to obesity in later life [128,129]. Neonates born to 25 (OH) D-deficient mothers exhibit larger superficial and deep SAT volumes, which bear metabolic similarities to adult VAT, despite having similar birth weights and VAT masses as neonates born to 25 (OH) D-sufficient mothers [130]. Similarly, animal studies have demonstrated that the offspring of mothers with VD deficiency display an elevated body weight, glucose intolerance, and an augmented risk of obesity in adulthood [131,132,133]. This could be attributed to VD deficiency during pregnancy causing epigenetic changes in the offspring, such as differential methylation in promoters and CpG islands, leading to the increased expression of VDR and PPAR-γ in adipose tissue. This, in turn, promotes the proliferation and differentiation of preadipocytes, ultimately contributing to the obese phenotype in the offspring [134,135]. Other potential influencing factors include adipocyte dysplasia, enhanced oxidation of adipose tissue, aberrant synthesis and secretion of adipokines, as well as inflammation [132].

In summary, the VDR plays a role in modulating adipogenesis and exhibits varying regulatory effects on adipocytes during diverse stages of differentiation (Figure 1). Such discrepancies between in vivo and in vitro studies might be partly explained by this, although further research is necessary to fully elucidate the underlying mechanisms involved.

## 6. Summary

In summary, there exists a strong association between VD levels and the onset and progression of obesity, as well as adipose tissue metabolic capacity and lipogenesis. Adipose tissue serves as the primary reservoir for VD, where VD stored in lipid droplets can be released into the bloodstream during lipolysis. The 1,25(OH)2D3/VDR complex and its subsequent signaling pathways possess the ability to directly govern adipogenesis, thus playing a significant role in the modulation of insulin sensitivity and inflammation within adipose tissue. Distinct adipose tissue types demonstrate variations in cellular composition, metabolic characteristics, extracellular matrix composition, and susceptibility to alterations in the internal milieu, which might mediate the differential effects of VD in different adipose depots. The accumulation of VAT stands as the most sensitive and crucial indicative factor for identifying the presence of VDD high-risk individuals. While certain observational studies fail to support the favorable effects of VD supplementation in ameliorating obesity and its associated complications, the consolidated evidence from existing research supports the advantageous role of VD/VDR in regulating adipose tissue health and preventing obesity.

## Figures and Tables

**Figure 1 nutrients-15-04832-f001:**
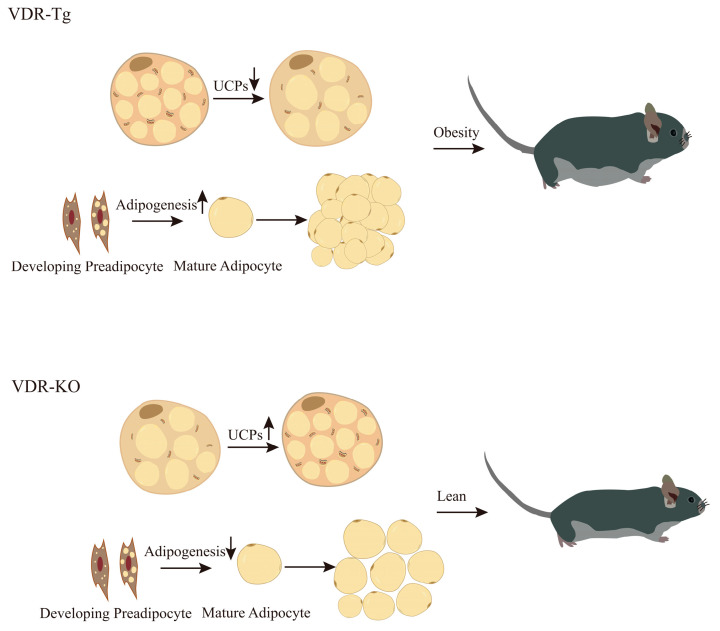
Role of VDR in the development of obesity. The presented figure depicts that the genetic manipulation of whole-body VDR, including knockout and overexpression, exerts notable effects on the browning phenomenon within white WAT as well as adipogenesis, consequently influencing the initiation and progression of obesity.

## Data Availability

Not applicable.

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
