# Peer review of "Interplay between Vitamin D and Adipose Tissue: Implications for Adipogenesis and Adipose Tissue Function"

_nutrients, 2023, doi:10.3390/nu15224832_

Round 1

Reviewer 1 Report

Comments and Suggestions for Authors

This review is well-written and of interest for readers of Nutrients. It brings updated information about the relationship between VD and adipose tissue function. It also provides an interesting presentation and thinking about the specific role of VAT and SAT in this relationship.

No major point has to be corrected but few little points have to be improved.

Ref 4 not appropriate since related to the action of VD on adipose tissue and not to adipose tissue and adipocyte

Ref 11 not appropriate please refer to a review depicting relationship between VD and obesity or adiposity for ex. Bennour et al Nutrients 2022

Please reformulate the sentence line 89 and 90. We understand that it is the VDBP that is stored in fat instead of VD.

Ref 39 and associated text in lines 124 to 126 are surprising here, since it deals with impact of VD on muscle function and not adipose tissue or a parameter linked to the previous sentence.

Line 136 and on many other lines, vitamin D instead of VD. Please homogenize.

Ref 45 not appropriate since it is related to inflammasome not adipose cell populations

Ref 63 not appropriate

Line 224 and 231 and 284, 1,25-dihydroxyvitamin D (1,25(OH)2D3) already defined previously. Please note that 1,25(OH)2D3 Is not written in a homogenous way throughout the text, please homogenize.

Line 225, ref 20 could be replace by ref 29.

Line 233, ref 85, please precise that such relationship between VDD and VDR expression has been describes in a specific context of bladder cancer cells.

The anti-inflammatory effect of VD has been depicted through its impact on cytokines and chemokines, but it is noteworthy that miRNA are also interesting targets for inflammation control, and that some miRNA are targeted by VD (ref Karkeni et al. Epigenetics 2018). Few words could be added.

In the paragraph related to the effect of VD on UCP1 expression (and the conclusion of this part, line 249 to 351), It is mandatory to present the article of Malloy et al. Mol Endocrinol 2013 who depicted the specific role of VDR, independently of VD on the regulation of UCP1. It is important since it decoupled the role of VD and the role of VDR regarding the regulation of UCP1. This remark is also important for comment on line 456 and further, where authors could also comment discrepancies between VD and VDR in terms of independent effect of VD and VDR.

Line 462 to 464 there are some “ “ which means that it is a citation ? Please add a reference in this case, or discard the “ “.

The authors should add the reference Seipelt et al.Faseb J 2020 to ref 132 and 133, since these authors also depicted an impact of VDD on adipose tissue metabolism of the offspring

Author Response

This review is well-written and of interest for readers of Nutrients. It brings updated information about the relationship between VD and adipose tissue function. It also provides an interesting presentation and thinking about the specific role of VAT and SAT in this relationship.

Dear reviewer, 

Thank you for your thoughtful and valuable comments on our manuscript, which we have made as feedback and revisions.

No major point has to be corrected but few little points have to be improved.

Ref 4 not appropriate since related to the action of VD on adipose tissue and not to adipose tissue and adipocyte

  1. The purpose of referencing reference 4 is to give those who are not familiar with this field some concepts of adipose tissue, to better understand the extensive discussion of white adipocytes in the following, and to pave the way for further exploration of the relationship between VD and white adipose tissue.

    Ref 11 not appropriate please refer to a review depicting relationship between VD and obesity or adiposity for ex. Bennour et al Nutrients 2022

  2. Reference 11 has been replaced by Vitamin D and Obesity/Adiposity-A Brief Overview of Recent Studies published by Bennour in 2022.

Please reformulate the sentence line 89 and 90. We understand that it is the VDBP that is stored in fat instead of VD.

  • Lines 89 and 90 describe how VD3 binds to VDBP and is transported to the liver for the next stage of the reaction.

Ref 39 and associated text in lines 124 to 126 are surprising here, since it deals with impact of VD on muscle function and not adipose tissue or a parameter linked to the previous sentence.

  • Reference 39 has been removed.

Line 136 and on many other lines, vitamin D instead of VD. Please homogenize.

  • The expression of Vitamin D in the text has been homogenized.

Ref 45 not appropriate since it is related to inflammasome not adipose cell populations

  • The purpose of quoting reference 45 here is to explain the complexity of adipose tissue composition and to prepare the ground for the following explanation of VD and adipose tissue inflammation.

Ref 63 not appropriate

  • Reference 63 has been modified, see lines 187 through 191 for details.

Line 224 and 231 and 284, 1,25-dihydroxyvitamin D (1,25(OH)2D3) already defined previously. Please note that 1,25(OH)2D3 Is not written in a homogenous way throughout the text, please homogenize.

  • The expression of 1,25(OH)2D3 in the article has been homogenized.

Line 225, ref 20 could be replace by ref 29.

  • Reference 20 has been replaced with reference 29, see line 219 for details.

Line 233, ref 85, please precise that such relationship between VDD and VDR expression has been describes in a specific context of bladder cancer cells.

  • The paragraph cited in reference 85 has been revised; for details, see lines 224 through 227.

The anti-inflammatory effect of VD has been depicted through its impact on cytokines and chemokines, but it is noteworthy that miRNA are also interesting targets for inflammation control, and that some miRNA are targeted by VD (ref Karkeni et al. Epigenetics 2018). Few words could be added.

In the paragraph related to the effect of VD on UCP1 expression (and the conclusion of this part, line 249 to 351), It is mandatory to present the article of Malloy et al. Mol Endocrinol 2013 who depicted the specific role of VDR, independently of VD on the regulation of UCP1. It is important since it decoupled the role of VD and the role of VDR regarding the regulation of UCP1. This remark is also important for comment on line 456 and further, where authors could also comment discrepancies between VD and VDR in terms of independent effect of VD and VDR.

Line 462 to 464 there are some “ “ which means that it is a citation ? Please add a reference in this case, or discard the “ “.

The authors should add the reference Seipelt et al.Faseb J 2020 to ref 132 and 133, since these authors also depicted an impact of VDD on adipose tissue metabolism of the offspring

  • References have been added to the corresponding paragraphs, see lines 468 to 470 for details.

Thank you again for your valuable comments on our shortcomings and wish you a pleasant life.

Shiqi Lu,11 Nov 2023

Reviewer 2 Report

Comments and Suggestions for Authors

This is a review article focused on the relationship between vitamin D and adipose tissue metabolism. Several issues associated with this topic are discussed in detail and a lot of data are presented. Nevertheless, I have some suggestions how the manuscript could be improved.

1)     Introduction: endocrine function of adipose tissue should be briefly discussed. In addition, the manuscript could benefit from addressing adipose tissue dysfunction in obesity and other metabolic diseases.

2)     Line 134: “in vivo studies” are mentioned two times.

3)     Line 145: “contraction” should better be replaced by: “compression”. Adipose tissue is not the contractile tissue and “contraction” is not appropriate.

4)     Lines 221-222: do percent contents of vitamin D metabolites in lipid droplets refer to whole-body content as 100% or whole content in the adipose tissue as 100%.

5)     When the effect of VD on different adipose tissue depots is presented, it would be important to clearly separate the results of in vitro and in vivo studies. It would be reasonable to include the table presenting data such as concentration/dose of VD used, experimental model (cells, adipose tissue depots) and the observed results.

6)     Authors used alternativelu “UCP” or “Ucp” for uncoupling proteins. The abbreviation system should be unified.

7) The topic has been previously covered in several review articles. It would be reasonable to specify what are the innovative aspects of this paper. In addition, the methods of literature search and analysis should be described.

Comments on the Quality of English Language

The manuscript requires moderate Englisch polishing.

Author Response

Dear reviewer,

This is a review article focused on the relationship between vitamin D and adipose tissue metabolism. Several issues associated with this topic are discussed in detail and a lot of data are presented. Nevertheless, I have some suggestions how the manuscript could be improved.

First of all, thank you very much for your careful comments on our manuscript. We have benefited a lot from the questions you raised. The following is my modification and response to your comments.

1)     Introduction: endocrine function of adipose tissue should be briefly discussed. In addition, the manuscript could benefit from addressing adipose tissue dysfunction in obesity and other metabolic diseases.

  • The purpose of referencing reference 4 is to give those who are not familiar with this field some concepts of adipose tissue, to better understand the extensive discussion of white adipocytes in the following, and to pave the way for further exploration of the relationship between VD and white adipose tissue.

2)     Line 134: “in vivo studies” are mentioned two times.

  •  The content of line 134 has been modified, see line 130 for details.

3)     Line 145: “contraction” should better be replaced by: “compression”. Adipose tissue is not the contractile tissue and “contraction” is not appropriate.

  • The content of line 145 has been modified, see line 140 for details.

4)     Lines 221-222: do percent contents of vitamin D metabolites in lipid droplets refer to whole-body content as 100% or whole content in the adipose tissue as 100%.

  • The data in lines 221 through 222 refer to whole-body VD levels; the corresponding paragraphs have been amended; see lines 215 through 216 for details.

5)     When the effect of VD on different adipose tissue depots is presented, it would be important to clearly separate the results of in vitro and in vivo studies. It would be reasonable to include the table presenting data such as concentration/dose of VD used, experimental model (cells, adipose tissue depots) and the observed results.

  • We think your suggestion is very good, but I am sorry that it is difficult for me to integrate them into a table and present them because of the heterogeneity of experimental types.

6)     Authors used alternativelu “UCP” or “Ucp” for uncoupling proteins. The abbreviation system should be unified.

  • The format of UCP has been homogenized.

7) The topic has been previously covered in several review articles. It would be reasonable to specify what are the innovative aspects of this paper. In addition, the methods of literature search and analysis should be described.

  • The innovation of this manuscript is to explore the effects of vitamin D on fat health in multiple dimensions, such as different types of adipocytes and adipocyte growth stages, maternal vitamin D deficiency and offspring, and to explain the related phenomena based on the latest research progress.
  • Because the search terms were not included in the manuscript considering that this manuscript is a review, the following is how we conducted the search.

PubMed and Google Scholar were used to search for relevant articles published between 1971 and 2023. The combination of search terms is as follows: ((vitamin D) OR (vitamin D deficiency) OR (vitamin D insufficiency) OR (vitamin D supplementation) OR (vitamin D) metabolism) OR (vitamin D action) OR (vitamin D receptor) OR (25(OH)D) OR (1,25(OH)2D) OR (obesity) OR (overweight) OR (fat mass) OR (abdominal obesity) OR (central obesity) OR (visceral adipose tissue) OR (subcutaneous adipose tissue) OR  (brown adipose tissue) OR (human adipose tissue) OR (preadipocyte) OR (adipocyte) OR (adipocyte differentiation) OR  (adipogenesis) OR (adipose tissue function) OR (adipose tissue metabolism) OR (lipolysis) OR (lipogenesis) OR (lipid  metabolism) OR (energy metabolism) OR (CYP2R1) OR (CYP27B1) OR (CYP24A1) OR (inflammation) OR (inflammatory cytokines)  OR (cytokine) OR (immune response) OR (adipokine) OR (adiponectin) OR (vitamin D supplementation) OR (NLRP3) OR  (caspase-1) OR (IL-1β). Clinical and preclinical studies involving human subjects. Included studies included randomized controlled trials, observational studies and meta-analyses. Studies with only abstract were excluded.

Thank you again for pointing out and guiding our problems. I hope you have a pleasant life.

Shiqi Lu, 11 Nov 2023

Round 2

Reviewer 2 Report

Comments and Suggestions for Authors

The manuscript has been revised according to the reviewers' comments. I have no further concerns.